# Differences in Preventive Behaviors of COVID-19 between Urban and Rural Residents: Lessons Learned from A Cross-Sectional Study in China

**DOI:** 10.3390/ijerph17124437

**Published:** 2020-06-20

**Authors:** Xuewei Chen, Hongliang Chen

**Affiliations:** 1School of Community Health Sciences, Counseling and Counseling Psychology, Oklahoma State University, Stillwater, OK 74078, USA; xuewei.chen@okstate.edu; 2College of Media and International Culture, Public Diplomacy and Strategic Communication Research Center, Zhejiang University, Hangzhou 310058, China

**Keywords:** COVID-19, rural-urban health disparities, critical heath literacy, information appraisal, theory of reasoned action, structural equation modeling

## Abstract

*Purpose:* The purpose of this study is to examine the differences in preventive behaviors of COVID-19 between urban and rural residents, as well as identify the factors that might contribute to such differences. *Methods:* Our online survey included 1591 participants from 31 provinces of China with 87% urban and 13% rural residents. We performed multiple linear regressions and path analysis to examine the relationship between rural status and behavioral intention, attitude, subjective norms, information appraisal, knowledge, variety of information source use, and preventive behaviors against COVID-19. *Findings:* Compared with urban residents, rural residents were less likely to perform preventive behaviors, more likely to hold a negative attitude toward the effectiveness of performing preventive behaviors, and more likely to have lower levels of information appraisal skills. We identified information appraisal as a significant factor that might contribute to the rural/urban differences in preventive behaviors against COVID-19 through attitude, subjective norms, and intention. We found no rural/urban differences in behavioral intention, subjective norms, knowledge about preventive behaviors, or the variety of interpersonal/media source use. *Conclusions:* As the first wave of the pandemic inundated urban areas, the current media coverage about COVID-19 prevention may not fully satisfy the specific needs of rural populations. Thus, rural residents were less likely to engage in a thoughtful process of information appraisal and adopt the appropriate preventive measures. Tailoring health messages to meet rural populations’ unique needs can be an effective strategy to promote preventive health behaviors against COVID-19.

## 1. Introduction

Studies documented that rural populations are facing health disparities due to multiple barriers such as lack of health care resources (e.g., transportation, health insurance, providers, and facilities), geographic distance, and lower socioeconomic status [1,2]. Compared with urban residents, rural residents have higher rates of morbidity and mortality from various diseases, including cancer and cardiovascular disease [3,4,5]. Rural populations also engage in preventive health behaviors less than urban populations. Preventive health behaviors refer to any activity undertaken by an individual who believes himself or herself to be healthy for the purpose of preventing disease [6]. For example, children living in rural areas consume less fruit and vegetable than their urban peers [7]; rural residents were less likely to wear sunscreen to prevent skin cancer than urban residents [8]; and women living in rural locations were less likely to receive mammography and Papanicolaou (Pap) smear screening to prevent cervical and breast cancer than their urban counterparts [9].

Rural residents are still encountering health disparities regarding infectious disease prevention and treatment. For instance, deaths from infectious diseases decreased by 18% in the United States between 1980 and 2014; however, rural counties did not experience the same improvements as their urban counterparts [10]. Similarly, in China, the levels of knowledge and awareness of the human immunodeficiency virus (HIV), tuberculosis (TB) and hepatitis B virus (HBV) are still low among rural residents [11]. Moreover, unique challenges (e.g., resource constraints and staff shortages in health care) are affecting rural areas’ ability to detect, respond, prevent, and control infectious disease outbreaks in both China and the U.S. [12,13].

The outbreak of COVID-19, an infectious disease, is currently causing a global public health crisis. The first case of COVID-19 was reported to the World Health Organization (WHO) on December 31, 2019 from Wuhan, China, and the outbreak was declared a Public Health Emergency of International Concern on January 30, 2020 [14]. As of May 2020, there were more than 4.2 million confirmed cases worldwide [15]. People with underlying medical conditions, including those with diabetes and chronic diseases (e.g., lung, kidney, and heart), might be at higher risk for severe illness from COVID-19 [16]. Public health researchers are concerned that rural communities might experience a worse situation related to the COVID-19 pandemic (e.g., greater mortality rates) than their urban and suburban counterparts due to the existing rural/urban health disparities [17,18,19,20]. For example, the higher rates of chronic diseases and less physical exercises impose higher risks of severe illness on rural cases [21]. Also, existing research documented that the escalation of COVID-19 spread is highly related to the transportation of people with no-to-mild symptoms—namely, those are unaware about the infection [22]. Therefore, health professionals recommend staying at home, social distancing, wearing facemasks, and frequent hand washing as effective containment measures. The promotion of these preventive behaviors is essential to slow down the spread of the virus during the outbreak [19,23]. The purpose of this study is to examine the differences in preventive behaviors of COVID-19 between urban and rural residents, as well as identify the factors that might contribute to such discrepancies. We hypothesize that rural residents are less likely to perform preventive health behaviors against COVID-19 compared with their urban counterparts.

## 2. Theoretical Framework

Research on social and behavioral sciences provides insights for effective responses to the COVID-19 pandemic [24]. Theory of reasoned action (TRA) proposes that an individual’s preventive behavior is a function of his or her intention to perform it, which is determined by an individual’s attitude and subjective norms towards a particular behavior [25,26]. Attitude comprises beliefs, values, and knowledge; subjective norms refer to the person’s perceptions about what important people want he or she to do with regard to the preventive behavior [25]. The framework of TRA proved to be effective in examining various preventive health behaviors such as condom use, physical activity, and diet control [27]. Grafting TRA to the context of COVID-19, we hypothesize that people’s behavioral intention is the main predictor of preventive behaviors, while attitude and subjective norms are two determinants of behavioral intention. We also hypothesize that rural/urban disparities in intention, attitude, and subjective norms lead to the differences in preventive behaviors.

Use of health information is crucial to personal and public health outcomes because it helps individuals accumulate knowledge and adopt healthier behavioral patterns [28]. Overwhelmed with information regarding COVID-19, it is challenging for individuals to evaluate the quality of related news and official recommendations [29]. Information appraisal refers to the critical analysis of health-related information, which is an important component of critical health literacy [30,31]. Information appraisal reflects the skills to apply health information to individual circumstances and process what a specific health behavior means in people’s “own world” [32,33]. Information appraisal skills are critical to health care consumers in the future. This is because of two major societal changes occurred recently that shaped the way people interact with health information: (1) the transition from a doctor-centered approach to the patient-centered one, and (2) the increasing volume and use of online health information [34]. The majority of related research focuses on functional health literacy (the ability to understand factual health information) [35,36]. Among the limited studies regarding information appraisal skills, researchers documented that information appraisal was associated with people’s vaccination attitude [37] and physical activity intention [38]. In the current study, we hypothesize information appraisal as a mediator contributing to rural/urban differences in preventive behaviors against COVID-19 through intention, attitude, and subjective norms. Moreover, previous study indicated that knowledge is another predictor of intention to adopt health promotion behaviors [39]. Individuals who are exposed to various sources of health information are likely to be knowledgeable about the health risks and benefits of preventive measures, which facilitates health decision-making and behavior changes [40]. Therefore, knowledge and variety of interpersonal/media source use may account for the rural/urban differences in preventive behaviors against COVID-19 through intention, attitude, and subjective norms.

Guided by the above theoretical concepts and existing literature, we examined the rural/urban differences in COVID-19 preventive behaviors, intention, attitude, subjective norms, knowledge, variety of interpersonal information source use (e.g., friends, health professionals), variety of media information source use (e.g., TV, websites), and information appraisal. We proposed a hypothesized path model testing the direct and indirect effects of rural status on preventive behaviors through intention, attitude, and subjective norms. We also examined the factors (i.e., knowledge, variety of interpersonal/media information source use, and information appraisal) that might contribute to the differences in preventive behaviors of COVID-19 between urban and rural residents.

## 3. Methods

### 3.1. Procedure and Participants

Data were drawn from a larger study designed to examine public risk communication combating COVID-19 in China. The online survey was conducted between January 31 and February 4, 2020, when COVD-19 began to spread nationally in China. We used SoJump to recruit participants (http://www.sojump.com). SoJump is one of the largest online survey providers in China that has over 2.6 million registered users with diverse sociodemographic characteristics [41,42]. SoJump used its internal record about its registered users to identify potential participants who were eligible for this study. SoJump then sent out study invitations to 1717 individuals, a randomly selected subset of the registered users. To be eligible for this study, participants had to be 16 years or older, living in mainland China (excluding Macau and Hong Kong), and be literate in Chinese. The Institutional Review Board at Zhejiang University approved the data collection protocol.

There were 1692 people (98.5% of invited respondents) who completed the survey. Each participant received a 15 RMB (approximate to 2 US dollars) e-gift card as incentive after completing the survey. Invalid responses were dropped because they met at least one of the following two priori criteria: (1) repetitive submission using the same IP address or (2) answered any of the three attention checkers incorrectly (e.g., “Survey validation item”, please select “some preparation”). The final sample included 1591 valid participants (50% male) from 31 provincial-level administrative units of China with 1381 (87%) urban and 210 (13%) rural residents. Participants’ ages ranged from 16 to 71 years (*M* = 31, *SD* = 9). Most respondents claimed some college education, including elementary school (0.1%), middle school (1.1%), high school (3.9%), professional school (2.26%), associate’s degree (13.6%), bachelor’s degree (69.3%), master’s degree (9.1%), and doctorate or postdoctoral degree (0.7%). The average monthly household income was between 8001 and 10,000 RMB; specific responses included no income (2.0%), 1000 and below (1.3%), 1001–3000 (4.7%), 3001–5000 (10.1%), 5001–8000 (16.3%), 8001–10,000 (14.6%), 10,001–15,000 (21.2%), 15,001–20,000 (15.7%), 20,001–50,000 (12.5%), and 50,001 and above (1.5%).

### 3.2. Measures

#### 3.2.1. Rural–Urban Residence

Participants self-reported their current geographic locations among six administrative categories: provincial-level municipalities, sub-provincial cities, prefecture-level cities, county-level cities, township-level divisions, and administrative villages. These administrative types of cities and towns have evolved over years since 1950s in response to national economic and social policies [43]. The four industrial giants of Shanghai (24.3 million population), Tianjin (11.6 million population), Beijing (21.5 million population), and Chongqing (30.5 million population), were designated as provincial-level municipalities, reporting directly to the central government. A sub-provincial city is governed by a province, but is administered independently in regard to economy and law. The population size of sub-provincial cities ranges from 3.5 million to 16.3 million. A prefecture-level city is a political subdivision roughly equivalent to a metropolitan area that ranks below a province but above a county. The average population size of prefecture-level cities is about 4 million. County-level cities have judicial but no legislative rights over their own local law and are usually governed by prefecture-level divisions. Township-level divisions contain village committees and villages. An administrative village is the last level of administrative division, underneath a township. As of January 2019, China has 15 sub-provincial cities, 294 prefecture-level cities, 387 county-level cities, more than 38,000 township-level divisions, and about 700,000 administrative villages. The classification of administrative divisions in China depends on multiple factors including policy, economy, culture, ethnicity, geography, population, and history [44]. Although there is no unified approach of distinguishing rural and urban areas in China [45], the first four categories are usually considered “urban” settlement and the last two categories are considered “rural” areas [43]. We dichotomized the geographic location variable into two categories: urban (i.e., provincial-level municipalities, sub-provincial cities, prefecture-level cities, and county-level cities) and rural (i.e., township-level divisions and administrative villages).

#### 3.2.2. Preventive Behaviors

We selected eight types of preventive behaviors from the COVID-19 prevention guidebook provided by WHO [46] and the Chinese Centers for Disease Control and Prevention [47]. Such behaviors included (1) wearing a mask when going out, (2) staying home as much as possible, (3) avoiding party gathering, (4) washing hands frequently, (5) avoiding public transportation, (6) trying to eat healthy and well-balanced meals, (7) getting plenty of sleep, and (8) exercising regularly. The aforementioned behaviors were measured with eight statements on a five-point Likert scale (1 = completely disagree, 2 = disagree, 3 = neither agree nor disagree, 4 = agree, 5 = completely agree). We calculated mean scores for these eight items, of which higher scores represented more engagement in preventive behaviors.

#### 3.2.3. Sociodemographic

Sociodemographic variables included age, sex, household monthly income in RMB (1 RMB = 0.14 US dollar) (0, ≤1k, 1k to ≤3k, 3k to ≤5k, 5k to ≤8k, 8k to ≤10k, 10k to ≤15k, 15k to ≤20k, 20k to ≤50k, above 50k), and education (elementary school, middle school, high school, professional school, associate degree, bachelor degree, master’s degree, and doctoral degree).

### 3.3. Potential Mediation Variables

#### 3.3.1. Behavioral Intention

We assessed participants’ intention to adopt preventive behaviors using a single item (“After knowing about the COVID-19 pandemic situation, I intend to take preventive behaviors.”) on a five-point Likert scale from completely disagree to completely agree (*M* = 4.21, *SD* = 0.63).

#### 3.3.2. Attitude

We also measured participants’ attitude about the effectiveness of performing preventive behaviors using a single item (i.e., “performing preventive behaviors might not effectively prevent getting the virus”) on a five-point Likert scale from completely disagree to completely agree. Responses were coded reversely so that the higher score indicated a more positive attitude (*M* = 3.68, *SD* = 1.03).

#### 3.3.3. Subjective Norms

We assessed subjective norms using eight items on a five-point Likert scale (Cronbach’s alpha = 0.84). One example item was “How prepared do your family or friends expect you to be for this pandemic?”. The options ranged from 1 (not at all) to 5 (a lot). We calculated the mean score of these eight items as our variable for subjective norms (*M* = 3.68, *SD* = 0.64).

#### 3.3.4. Knowledge about Preventive Behaviors

Knowledge about preventive behaviors was assessed using a single item (“I do not know what to do for preventive behaviors”) on a five-point Likert scale from completely disagree to completely agree. After the reverse coding, the higher score indicated higher knowledge (*M* = 4.00, *SD* = 0.80).

#### 3.3.5. Variety of Interpersonal Information Source

Participants were asked how frequently (“never”, “occasionally”, “sometimes”, “often”, or “very often”) they used interpersonal sources for information about COVID-19. The interpersonal sources included six items: (1) family members, (2) friends, (3) colleagues/classmates, (4) health professionals, (5) community workers, and (6) others. For each source, we dichotomized those who chose “often” and “always” as frequent users (coded as 1) and the rest as nonfrequent users (coded as 0). We focused on variety of information source use because checking various sources to confirm information and address discrepancies is a recommended strategy to ensure using accurate information and make appropriate health decision [48]. The sum score of the interpersonal sources was calculated to represent the level of interpersonal source variety (*M* = 1.93, *SD* = 1.33). The higher score indicated that the individual was a frequent user of more various types of source.

#### 3.3.6. Variety of Media Information Source

The media sources included eleven items: (1) newspapers/magazines, (2) TV, (3) radio, (4) cellphone text messages, (5) web portals (e.g., www.163.com, www.tencent.com), (6) social media (e.g., weibo, wechat), (7) news websites (e.g., China Daily, Toutiao), (8) video-sharing social networking service (e.g., TikTok, Pear Video), (9) online Q&A platforms (e.g., Zhihu), (10) search engines (e.g., Baidu), and (11) online learning platform (e.g., Xuexi Qiangguo). Similarly, we recoded the items and the sum score of the media sources was calculated to represent the level of media source variety (*M* = 5.57, *SD* = 2.21).

#### 3.3.7. Information Appraisal

We examined how people process COVID-19 information using six items on a five-point Likert scale (Cronbach’s alpha = 0.70). These items were adapted from earlier research [49,50]. These six items included “I tried to relate the information to my own personal experiences”, “I thought about the importance of this information to my daily life”, “I browsed the COVID-19 news with no specific focus”, “I only paid attention to a few piece of information”, “I did not think carefully about the point of view in the information”, and “I did not spend much time thinking about the information”. The last four items were coded reversely. We calculated a mean score for these six items. Higher score indicated a higher level of appraisal skills related to COVID-19 information they received from various sources (*M* = 3.93, *SD* = 0.57).

## 4. Data Analysis

### 4.1. Simple Linear Regressions

We performed simple linear regressions (unadjusted models) to examine the association between rural status and (1) preventive behaviors, (2) behavioral intention, (3) attitude, (4) subjective norms, (5) preventive behavior knowledge, (6) interpersonal source variety, (7) media source variety, and (8) information appraisal.

### 4.2. Multiple Linear Regressions

We also performed multiple linear regressions (adjusted models), controlling for demographic characteristics (i.e., age, sex, education, and income), to examine the relationship between rural status and the above eight variables.

### 4.3. Path Analysis

We then performed path analysis to test the indirect effects of rural status on preventive behaviors through the above seven potential mediators. We treated rural status as an exogenous variable and added the demographic control variables into the path model as exogenous variables. When evaluating how well a specific model fits the data, we used the following model fit indices: the model Chi-square value (χ^2^), the root mean square error of approximation (RMSEA), the comparative fit index (CFI), the Tucker−Lewis index (TLI), and the standardized root mean squared residual (SRMR). The model is considered a “good fit” when the χ^2^
*p*-value > 0.05, RMSEA < 0.06, CFI > 0.95, TLI > 0.95, and SRMR < 0.05 [51]. We followed modification indices to add suggested paths that can be theoretically justified to achieve good model fit. We used Stata 16 for data analysis and set the significance level at α = 0.05.

## 5. Results

### 5.1. Simple Linear Regressions

The results of the unadjusted models indicated that rural status was associated with lower preventive behaviors (b = −0.15, *p* < 0.001), lower intention (b = −0.19, *p* < 0.001), more negative attitude (b = −0.29, *p* < 0.001), lower knowledge (b = −0.18, *p* < 0.001), and lower information appraisal skills (b = −0.17, *p* < 0.001). We found no rural/urban differences in subjective norms or variety of interpersonal/media source use in the unadjusted models.

### 5.2. Multiple Linear Regressions

With regard to adjusted models when holding age, sex, education, and income constant (as shown in Table 1), compared with urban residents, rural residents were less likely to perform preventive behaviors, hold a more negative attitude, and have lower levels of information appraisal skills. Older respondents reported more engagement in preventive behaviors, greater intention, more positive attitude toward the effectiveness of performing preventive behaviors, and a greater variety of interpersonal source use. Higher education was associated with increased knowledge about preventive behaviors, greater variety of interpersonal source use, and higher information appraisal. Respondents with higher income reported more preventive behaviors, positive attitude, knowledge, and information appraisal. We found no rural/urban differences in behavioral intention, subjective norms, knowledge, or variety of interpersonal/media source use in the adjusted models.

### 5.3. Path Analysis

We tested four path models where one of the four potential mediators were included in each model (i.e., preventive behavior knowledge, variety of interpersonal source use, variety of media source use, and information appraisal) as the mediator between rural status and behavior, intention, attitude, and subjective norms. Based on the modification indices, we added one path from age to intention to improve the model fit. The model with information appraisal as the mediator exhibited good fit: χ^2^(10) = 17.45, *p* = 0.065, RMSEA = 0.022, CFI = 0.992, TLI = 0.973, SRMR = 0.015. About 20% of the variance in preventive behaviors can be explained/predicted by our model (R^2^ = 0.20). The other three models exhibited poor fit.

#### 5.3.1. Direct Effect

As shown in Figure 1 and Table 2, there was a direct effect of rural status on information appraisal (b = −0.10, *p* = 0.028), as well as direct effects of information appraisal on attitude (b = 0.45, *p* < 0.001), subjective norms (b = 0.29, *p* < 0.001), intention (b = 0.27, *p* < 0.001), and preventive behaviors (b = 0.13, *p* < 0.001). The direct effects of intention (b = 0.08, *p* < 0.001), attitude (b = 0.05, *p* < 0.001), subjective norms (b = 0.11, *p* < 0.001), and information appraisal (b = 0.13, *p* < 0.001) on preventive behaviors were also significant. In addition, there were direct effects of attitude (b = 0.05, *p* < 0.001) and subjective norms (b = 0.24, *p* < 0.001) on intention.

#### 5.3.2. Indirect Effect

Although the direct effect was nonsignificant, the indirect effects of rural status on preventive behaviors through information appraisal, attitude, subjective norms, and intention were significant (b = −0.03, *p* = 0.008). There were indirect effects of information appraisal on preventive behaviors through attitude, subjective norms, and intention (b = 0.08, *p* < 0.001). The indirect effects of attitude (b = 0.004, *p* = 0.004) and subjective norms (b = 0.02, *p* < 0.001) on preventive behaviors through intention were also significant.

#### 5.3.3. Total Effect

The total effects of rural status (b = −0.07, *p* = 0.029), attitude (b = 0.06, *p* < 0.001), intention (b = 0.08, *p* < 0.001), subjective norms (b = 0.13, *p* < 0.001), and information appraisal (b = 0.22, *p* < 0.001) on preventive behaviors were all significant. The total effects of rural status on intention (b = −0.04, *p* = 0.012), attitude (b = −0.18, *p* = 0.026), subjective norms (b = −0.03, *p* = 0.031), and information appraisal (b = −0.10, *p* = 0.028) were all significant as well.

## 6. Discussion

We explored the mechanism underlying the urban−rural differences in COVID-19 preventive behaviors. As we found, rural residents were less likely to engage in preventive behaviors, reported less positive attitude toward the effectiveness of performing preventive behaviors, and had lower levels of information appraisal skills. These findings were consistent with previous studies unveiling rural/urban health disparities in other preventive behaviors, such as wearing sunscreen [8] and receiving preventive care services including cancer screening [9] and influenza vaccinations [52]. Similarly, prior studies found that rural women were more likely to have a negative attitude about breast cancer and possess less positive attitude toward mammography screening compared with their urban counterparts [53,54]. We did not find rural/ urban differences in knowledge about preventive behaviors or interpersonal/media source variety.

Compared with urban residents, rural residents performed fewer preventive behaviors, held more negative attitude, and had poorer information appraisal skills, even after controlling for demographic characteristics. In the path models, we identified information appraisal as a significant factor that might contribute to the rural/urban differences in preventive behaviors through the mediation of attitude, subjective norms, and intention. Rural residents reported lower levels of information appraisal skills than their urban counterparts. In other words, rural residents were less likely to evaluate the relevance or salience of the information. Next, the poor information appraisal skill was associated with lower likelihood of holding a positive attitude about preventive behaviors, lower intention to adopt recommended behaviors, and lower level of subject norms, which lead to less engagement in preventive behaviors among rural residents. Similarly, a previous study found that those who paid more attention to H1N1 news were more likely to adopt preventive behaviors to protect themselves from influenza infection [55]. Our findings indicated that preventive behavior knowledge, variety of interpersonal source use, and variety of media source use were not significant factors accounting for the rural/urban differences in preventive behaviors through the mediation of attitude, subjective norms, and intention. These nonsignificant findings might be due to the nonsignificant associations between rural status and these factors. One possible direction for future study is to clarify this mechanism.

Next, we found that rural residents reported lower levels of information appraisal skills than their urban counterparts, controlling for demographic characteristics. One possible explanation is that the current media coverage about COVID-19 prevention mostly focuses on large urban cities with high population density, which might not fully satisfy the specific needs of rural populations. Thus, rural residents might not be strongly motivated to engage in a thoughtful process of information appraisal and adopt the appropriate preventive measures. Tailoring health messages to meet a person’s individual needs might be an effective strategy to promote preventive health behaviors against COVID-19 among rural audiences. Tailored health communication has been used to enhance information appraisal, increase motivation to process information, and promote behavioral change [56]. Rural residents have a strong sense of community and resilience [57,58]. Therefore, calling upon rural residents’ sense of community and highlighting how their actions can protect their neighbors and local economy could be another effective messaging strategy to promote preventive behaviors against COVID-19 in rural areas [19]. Another possible explanation for the rural/urban disparities in information appraisal skills is that rural residents might have less experience of internet use than urban residents, which inhibits them to conduct online health information searches. Incorporating information literacy education into the national health literacy promotion program can be an effective strategy to reduce this disparity in rural China [59]. Moreover, previous studies indicated that rural residents are more likely to rely on nurse practitioners and local health departments as usual sources of health information compared with their urban counterparts [28,60]. These two sources would be pivotal to disseminate reliable information about COVID-19 in rural areas.

Additionally, we found that compared with people with older age, younger individuals reported fewer preventive behaviors, lower intention to do so, and they were less likely to hold a positive attitude to behavioral change. This might relate to the rumor that older people are the only ones at risk for COVID-19. The fact is that older adults and people with existing medical conditions are at higher risk of getting the virus, but anyone can become sick [61]. Fake news and misinformation on social media is a problem prevailing in rural China during this pandemic [22]. Similarly, misinformation is a challenge to preventive medicine and public health in the United States [62]. We also found that lower income or/and education were associated with lower levels of behavioral performance, positive attitude, and knowledge related to COVID-19 preventive behaviors. Previous studies found that vulnerable populations are more likely to use and trust health information from social media where information accuracy and quality are questionable [63,64]. Public health efforts should be made to help the public better identify the rumors and misinformation related to COVID-19 pandemic. For example, creating easy-to-understand messages through the official social media accounts of government and health organizations can be an effective strategy to reach the rural communities.

In the unadjusted regression models, we found that rural residents had lower intention and knowledge than urban residents. However, holding age, sex, education, and income constant, the differences in intention and knowledge became nonsignificant. Our findings indicate that differences in socioeconomic factors between rural/urban residents are likely explanations for why rural/urban differences in behavioral intention and knowledge are observed. Previous studies documented that individuals with lower incomes and educational attainment reported lower intention to engage in preventive health behaviors [65] and limited knowledge about health risks [66]. The disadvantages in household income and education among rural residents in China [67] could yield the rural/urban health disparities in the context of COVID-19 pandemic.

Although the adjusted regression analysis did not indicate significant rural/urban differences in intention or subjective norms, our path model demonstrated that the total effects of attitude, subjective norms and intention on preventive behaviors were significant. Our path model results confirmed the framework of TRA, where attitude, subjective norms, and intention predict behaviors [25,26]. To curb the pandemic, it is important to increase people’s positive attitude toward preventive behaviors (e.g., social distancing, hand washing, and facemask wearing) and raise their normative beliefs that the preventive measure is a must-do to protect other community members.

## 7. Limitations

The cross-sectional design of the study mitigates our ability to infer causal relationships and we cannot rule out the possibility of reverse directions in the model. Studies conducting data collection at multiple time points of the pandemic could yield different results. In addition, secondary data analysis limited our measures and variable choices. Although the administrative divisions are commonly used to classify rural and urban status, there is lack of a unified approach to urban and rural classifications due to the rapid urbanization in China. Different classification could produce different results. Also, our online survey study design excluded individuals who have no access to internet, especially rural residents. Thus, our sample cannot be considered representative of the Chinese population, which might generate biased results. The rural residents recruited in our study might have higher social economic status than other rural residents because our rural sample have regular access to internet. The rural/urban disparities might be larger in reality. Last, our list of media sources is inclusive but not exhaustive. The landscape of media industry is unique in China, thereby our findings related to information source likely do not generalize to other countries.

## 8. Conclusions

Our study contributes to a body of evidence not only identifying the rural/urban differences in preventive behaviors against COVID-19 but also demonstrating that information appraisal is an important component associated with such urgent rural/urban health disparity during this pandemic. As the first wave of the pandemic inundated urban areas, the needs of rural populations are likely to be underrepresented in media. The ignorance and lack of awareness imposed greater risks of COVID-19 on rural communities. Public health efforts should be made to tailor COVID-19 information targeting rural populations.

## Figures and Tables

**Figure 1 ijerph-17-04437-f001:**
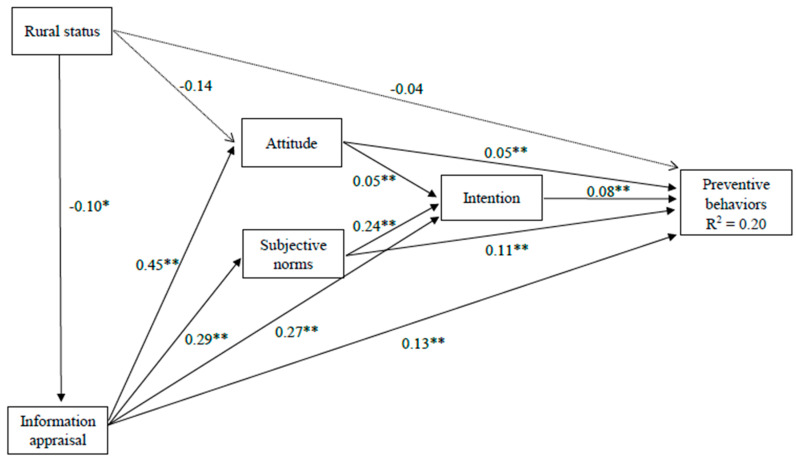
Path model diagram. * *p* < 0.05, ** *p* < 0.01.

**Table 1 ijerph-17-04437-t001:** Regression coefficients of rural/urban differences, controlling for sociodemographic.

	Rural Status	Age	Sex	Education	Income
Preventive behaviors	−0.07 *	0.01 **	0.03	0.02	0.01 *
Behavioral intention	−0.10	0.01 **	0.01	0.03	0.02
Attitude	−0.18 *	0.01 **	0.02	−0.03	0.03 *
Subjective norms	−0.02	−0.00	0.04	0.02	0.01
Knowledge	−0.06	−0.00	0.00	0.07 **	0.04 **
Interpersonal source variety	0.05	0.01 *	0.03	0.13 **	0.03
Media source variety	−0.03	−0.00	0.10	−0.01	−0.04
Information appraisal	−0.10 *	0.00	0.04	0.05 **	0.02 *

Note. * indicates *p* < 0.05; ** indicates *p* < 0.01.

**Table 2 ijerph-17-04437-t002:** Path analysis results.

	Independent Variable	b	SE	95% CI	*p*
**Direct Effect**					
Preventive behaviors	Rural status	−0.04	0.03	−0.10, 0.02	0.178
	Attitude	0.05	0.01	0.03, 0.07	<0.001 **
	Intention	0.08	0.02	0.05, 0.11	<0.001 **
	Subjective norms	0.11	0.02	0.08, 0.14	<0.001 **
	Information appraisal	0.13	0.02	0.10, 0.17	<0.001 **
Intention	Attitude	0.05	0.01	0.02, 0.08	<0.001 **
	Subjective norms	0.24	0.02	0.19, 0.28	<0.001 **
	Information appraisal	0.27	0.03	0.22, 0.32	<0.001 **
Attitude	Rural status	−0.14	0.08	−0.29, 0.02	0.083
	Information appraisal	0.45	0.04	0.36, 0.53	<0.001 **
Subject norms	Information appraisal	0.29	0.03	0.23, 0.34	<0.001 **
Information appraisal	Rural status	−0.10	0.05	−0.19, −0.01	0.028 *
**Indirect Effect**					
Preventive behaviors	Rural status	−0.03	0.01	−0.05, −0.01	0.008 **
	Information appraisal	0.08	0.01	0.07, 0.10	<.001 **
	Attitude	0.004	0.001	0.001, 0.007	0.004 **
	Subjective norms	0.02	0.004	0.01, 0.03	<.001 **
Intention	Rural status	−0.04	0.02	−0.08, −0.01	0.012 *
	Information appraisal	0.09	0.01	0.07, 0.11	<0.001 **
Attitude	Rural status	−0.04	0.02	−0.09, −0.00	0.032 *
Subjective norms	Rural status	−0.03	0.01	−0.05, −0.00	0.031 *
**Total Effect**					
Preventive behaviors	Rural status	−0.07	0.03	−0.13, −0.01	0.029 *
	Attitude	0.06	0.01	0.04, 0.08	<0.001 **
	Intention	0.08	0.02	0.05, 0.11	<0.001 **
	Subjective norms	0.13	0.02	0.10, 0.16	<0.001 **
	Information appraisal	0.22	0.02	0.18, 0.25	<0.001 **
Intention	Rural status	−0.04	0.02	−0.08, −0.01	0.012 *
	Information appraisal	0.36	0.03	0.31, 0.41	<0.001 **
	Attitude	0.05	0.01	0.02, 0.08	<0.001 **
	Subjective norms	0.24	0.02	0.19, 0.28	<0.001 **
Attitude	Rural status	−0.18	0.08	−0.34, −0.02	0.026 *
	Information appraisal	0.45	0.04	0.36, 0.53	<0.001 **
Subjective norms	Rural status	−0.03	0.01	−0.05, −0.00	0.031 *
	Information appraisal	0.29	0.03	0.23, 0.34	<0.001 **
Information appraisal	Rural status	−0.10	0.05	−0.19, −0.01	0.028 *

Note. b = regression coefficient; SE = standard error; CI = confidence interval; * indicates *p* < 0.05; ** indicates *p* < 0.01.

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
