# Peer review of "Differences in Preventive Behaviors of COVID-19 between Urban and Rural Residents: Lessons Learned from A Cross-Sectional Study in China"

_ijerph, 2020, doi:10.3390/ijerph17124437_

Round 1

Reviewer 1 Report

This is an interesting paper on an important topic: the differences in COVID-19 preventive behaviors between urban and rural residents and the potential mechanisms. I have some concerns with the paper's review of theory in developing its proposed path model and the method it adopted.

First, given the complexity of the design mentioned in the method section of the paper (the paper identified 6 potential mediators). The theoretical framework section of the manuscript is relative thin. While the authors emphasized that their proposed model is based on the Theory of Reasoned Action, many factors measured in the proposed model are actually not from the TRA model. Thus, without explaining in detail how factors such as information appraisal and information source type/variety can be integrated with the TRA model, it is unclear to readers regarding the theoretical links between TRA and other factors in the proposed model. Thus, it is ambiguous why these factors are selected and emphasized in this study.

Second, the paper claimed to center on the differences in preventive behaviors between rural and urban residents. However, although 13% of the sample (210 respondents) were classified as rural residents in the study, it is unclear whether these respondents are representative of the general rural residents in China because these respondents were recruited through an online survey portal. Thus, the survey is biased toward rural people who have access to internet and who are registered users of this online survey portal. Thus, it would be helpful to provide demographics information of both the rural and urban respondents and compare them to census data to see the degree to which the sample is representative of urban and rural residents in China. 

Third, while it is reasonable for the study to control for demographics variables (age, gender, education, income) when investigating rural-urban differences, the study failed to explain what really accounts for the observed difference in preventive behaviors between rural and urban residents. We know it is not due to differences in age, gender, education, and income between rural and urban residents as these variables were controlled. But then, what is at the root of the rural-urban difference in China? It would be helpful to discuss this in both the theory and the discussion sections.

Fourth, in the regression model, the authors found that rural status is a significant predictor of preventive behaviors but not behavioral intentions. It is a bit confusing given that behavioral intentions and behaviors should be highly correlated and thus, it would be helpful for authors to explain this finding further in the discussion section.

Fifth, in the path model, the authors argued that they selected mediating factors based on statistical significance. Information appraisal was included in the model because there was a significant relationship between rural status and information appraisal while other mediators were left out of the path model. Path models should be theory driven. Using statistical significance as the standard to decide what should be included and excluded from the path model may not be justified since it is too post hoc.

Sixth, authors did not mention how they revised the path model to achieve good model fit. 

Finally, as the authors acknowledged in the limitation section, the study adopted a cross-sectional design. Thus, it is ambiguous regarding the validity of those three-step and four-step mediation paths in the proposed model. It might be helpful to carefully justify these mediation paths using relevant literature and theories. Without a longitudinal design or an experimental design, the causal directions are unclear for some of the mediation paths in the proposed model.

Author Response

Dear Reviewer 1,

We would like to take this opportunity to thank you for the constructive feedback and the opportunity to make revisions to our manuscript. The comments were very helpful as we revised sections of our paper and has resulted in a stronger contribution to the literature. Below, we address each comment in turn, and indicate where in the manuscript text to find specific revisions and newly added information (with tracking changes).

Point 1:  First, given the complexity of the design mentioned in the method section of the paper (the paper identified 6 potential mediators). The theoretical framework section of the manuscript is relative thin. While the authors emphasized that their proposed model is based on the Theory of Reasoned Action, many factors measured in the proposed model are actually not from the TRA model. Thus, without explaining in detail how factors such as information appraisal and information source type/variety can be integrated with the TRA model, it is unclear to readers regarding the theoretical links between TRA and other factors in the proposed model. Thus, it is ambiguous why these factors are selected and emphasized in this study.

RESPONSE 1: We appreciate the reviewer’s suggestion. We strengthened the Theoretical Framework on pages 2-3.

Point 2: Second, the paper claimed to center on the differences in preventive behaviors between rural and urban residents. However, although 13% of the sample (210 respondents) were classified as rural residents in the study, it is unclear whether these respondents are representative of the general rural residents in China because these respondents were recruited through an online survey portal. Thus, the survey is biased toward rural people who have access to internet and who are registered users of this online survey portal. Thus, it would be helpful to provide demographics information of both the rural and urban respondents and compare them to census data to see the degree to which the sample is representative of urban and rural residents in China. 

RESPONSE 2: Rural population in China was reported at 40.85% in 2018. We agree with the reviewer’s point that our study might generate biased results. We added in our limitation section, “Also, our online survey study design excluded individuals who have no access to internet, especially rural residents. Thus, our sample cannot be considered representative of the Chinese population, which might generate biased results. The rural residents recruited in our study might have higher social economic status than other rural residents because our rural sample have regular access to internet. The rural/urban disparities might be larger in the population.”

Point 3: Third, while it is reasonable for the study to control for demographics variables (age, gender, education, income) when investigating rural-urban differences, the study failed to explain what really accounts for the observed difference in preventive behaviors between rural and urban residents. We know it is not due to differences in age, gender, education, and income between rural and urban residents as these variables were controlled. But then, what is at the root of the rural-urban difference in China? It would be helpful to discuss this in both the theory and the discussion sections.

RESPONSE 3: We added performing simple linear regressions (unadjusted models) to examine the association between rural status and (1) preventive behaviors, (2) behavioral intention, (3) attitude, (4) subjective norms, (5) preventive behavior knowledge, (6) interpersonal source variety, (7) media source variety, and (8) information appraisal. The comparisons between the results of unadjusted regression models and adjusted regression models (where holding age, sex, education, and income constant) indicated that these socioeconomic characteristics are likely explanations for why rural/urban differences in intention and knowledge are observed. After controlling age, sex, education, and income, the rural/urban differences in preventive behaviors still exist, which indicated that besides socioeconomic characteristics, there might be other factors contributing to such rural/urban disparities. Therefore, we tested the path model adding the potential mediators. We expanded it in the Theoretical Framework (p.2-3), Data Analysis (p.6), Results (p.6), and Discussion (p.11).

Point 4: Fourth, in the regression model, the authors found that rural status is a significant predictor of preventive behaviors but not behavioral intentions. It is a bit confusing given that behavioral intentions and behaviors should be highly correlated and thus, it would be helpful for authors to explain this finding further in the discussion section.

RESPONSE 4: In the unadjusted regression models, we found that rural residents had lower intention than urban residents. However, holding age, sex, education, and income constant rendered the differences in intention non-significant. Our results indicate that socioeconomic characteristics such as education and income are likely explanations for why rural/urban differences in intention are observed. We added the information related to the unadjusted regression models in Methods (p.6) and Results (p.6) and expanded the Discussion (p.11).

Point 5: Fifth, in the path model, the authors argued that they selected mediating factors based on statistical significance. Information appraisal was included in the model because there was a significant relationship between rural status and information appraisal while other mediators were left out of the path model. Path models should be theory driven. Using statistical significance as the standard to decide what should be included and excluded from the path model may not be justified since it is too post hoc.

RESPONSE 5: We tested four path models where including one of the four potential mediators in each model (i.e., preventive behavior knowledge, interpersonal source variety, media source variety, and information appraisal) as the mediator between rural status and behavior, intention, attitude, and subjective norms. These four potential mediators were selected based on theories and previous studies. The model with information appraisal as the mediator exhibited good fit; however, the rest three models exhibited poor fit. We clarified this in the Theoretical Framework (p.2-3), Results (p.7), and Discussion (p.10-11).

Point 6: Sixth, authors did not mention how they revised the path model to achieve good model fit. 

RESPONSE 6: Based on the modification indices, we added one path from age to intention to improve the model fit. We added this information in Methods (p.6) and Results (p.7). We also discussed the association between age, intention, attitude, and behavior in the Discussion (p.11).

Point 7: Finally, as the authors acknowledged in the limitation section, the study adopted a cross-sectional design. Thus, it is ambiguous regarding the validity of those three-step and four-step mediation paths in the proposed model. It might be helpful to carefully justify these mediation paths using relevant literature and theories. Without a longitudinal design or an experimental design, the causal directions are unclear for some of the mediation paths in the proposed model.

RESPONSE 7: We agree with the reviewer that without a longitudinal or experimental design, we cannot derive causal relationships. We admit that the cross-sectional design of the study mitigates our ability to infer causal relationships and we cannot rule out the possibility of reverse directions in the model (p.11-12).

Reviewer 2 Report

Thank you for the opportunity to review this manuscript. The article addresses an important topic and the results about preventive behaviors during this global pandemic are important to researchers and lay populations. I have some concerns about the manuscript and offer several suggestions for improving the overall quality of the writing as well as the introduction, methods, discussion sections.

Abstract

Line 21: Please clarify if rural residents were more or less likely to have, “lower levels of information appraisal skills.”

Line 24: Please change the word, “difference” to “differences.”

Introduction

Lines 39-40: I recommend revising this sentence to include the word, “positive” to describe health behaviors.

Line 45: please change “disparity” to “disparities.”

Lines 50-52: Sentence is missing a verb – consider including the word, “ability.” Full sentence would read, “Moreover, unique challenges (e.g., resource constraints and staff shortages in healthcare) are affecting rural areas’ ability to detect, respond, prevent, and control infectious disease outbreaks [11, 12]. Also, please specify if these unique challenges occur in the US, China, or both.

Line 53: COVID-19 is not a disease outbreak, rather there is a global public health crisis due to the outbreak of COVID-19. Revise sentence to reflect this (e.g., “The outbreak of COVID-19, an infectious disease, is currently causing a global public health crisis.”).

Lines 57-58: Please revise sentence for parallel structure by giving examples of lung, kidney, and heart diseases.

Line 61: The word, “chronical” should be “chronic.”

Theoretical framework: The section of the use of information and information appraisal needs strengthening. I recommend differentiating between media and interpersonal sources of information as described in the methods and adding examples. Please also add the potential implications associated with/for individuals who are capable of critically evaluating or appraising health information.

Methods

Please add the following to the manuscript: The number and percentage of SoJump users who were randomly selected and contacted to participate in the study. The overall survey completion rate.  

Regarding the priori criteria for dropping responses – were participants dropped for their failure to complete any of the three attention checkers or for failing any of the three attention checkers? Please clarify.

Please include a brief description of each of the six administrative categories. An “i.e.,” with the population determination for each category is fine (e.g., “i.e., over 10 million people”). Please also include a sentence describing the final numerical, population determinants for urban and rural. *I understand the classification system is listed later in the paper as a potential limitation, but it is important to explain what was used for this study in the manuscript.  

Good idea to include the actual, WHO issued preventive behaviors in the survey.

For each potential mediation variable, please include the total number of items used or specify if the measure was a single item.

What were the five response options for the variety of interpersonal information sources? Never to very often is listed in the measure, but later, the authors indicate that “often” and “always” were combined to reflect frequent users. Please clarify and add this information to the manuscript.

What was the rationale for dichotomizing the variety of interpersonal information sources?

How were media sources evaluated by participants? Please clarify and add that information to the manuscript.

Discussion: I would like to see more of an explanation as to why rural residents have poorer information appraisal skills than urban residents and their implications. In other words, I’m asking the authors to discuss potential reasons for the discrepancies in information appraisal between rural and urban residents and their implications for rural communities during the pandemic.

Author Response

Dear Reviewer 2,

We would like to take this opportunity to thank you for the constructive feedback and the opportunity to make revisions to our manuscript. The comments were very helpful as we revised sections of our paper and has resulted in a stronger contribution to the literature. Below, we address each comment in turn, and indicate where in the manuscript text to find specific revisions and newly added information (with tracking changes).

Point 1: Line 21: Please clarify if rural residents were more or less likely to have, “lower levels of information appraisal skills.”

RESPONSE 1: We clarified that rural residents were more likely to have lower levels of information appraisal skills.

Point 2: Line 24: Please change the word, “difference” to “differences.”

RESPONSE 2: Thanks for pointing this out. We made that change. We also made the change throughout the manuscript for consistency.

Point 3: Lines 39-40: I recommend revising this sentence to include the word, “positive” to describe health behaviors.

RESPONSE 3: We clarified the term “preventive health behaviors” and added a definition of it.

Point 4: Line 45: please change “disparity” to “disparities.”

RESPONSE 4: We changed it.

Point 5: Lines 50-52: Sentence is missing a verb – consider including the word, “ability.” Full sentence would read, “Moreover, unique challenges (e.g., resource constraints and staff shortages in healthcare) are affecting rural areas’ ability to detect, respond, prevent, and control infectious disease outbreaks [11, 12]. Also, please specify if these unique challenges occur in the US, China, or both.

RESPONSE 5: We made that change and clarified that these challenges occur in both China and the U.S.

Point 6: Line 53: COVID-19 is not a disease outbreak, rather there is a global public health crisis due to the outbreak of COVID-19. Revise sentence to reflect this (e.g., “The outbreak of COVID-19, an infectious disease, is currently causing a global public health crisis.”).

RESPONSE 6: We appreciate the reviewer’s suggestion and made the change accordingly.

Point 7: Lines 57-58: Please revise sentence for parallel structure by giving examples of lung, kidney, and heart diseases.

RESPONSE 7: We revised that sentence.

Point 8: Line 61: The word, “chronical” should be “chronic.”

RESPONSE 8: We corrected this typo.

Point 9: Theoretical framework: The section of the use of information and information appraisal needs strengthening. I recommend differentiating between media and interpersonal sources of information as described in the methods and adding examples. Please also add the potential implications associated with/for individuals who are capable of critically evaluating or appraising health information.

RESPONSE 9: We expanded the Theoretical framework on pages 2-3.

Methods

Point 10: Please add the following to the manuscript: The number and percentage of SoJump users who were randomly selected and contacted to participate in the study. The overall survey completion rate.  

RESPONSE 10: We added this information on page 3.

Point 11: Regarding the priori criteria for dropping responses – were participants dropped for their failure to complete any of the three attention checkers or for failing any of the three attention checkers? Please clarify.

RESPONSE 11: We clarified it into “answered any of the three attention checkers incorrectly”.

Point 12: Please include a brief description of each of the six administrative categories. An “i.e.,” with the population determination for each category is fine (e.g., “i.e., over 10 million people”). Please also include a sentence describing the final numerical, population determinants for urban and rural. *I understand the classification system is listed later in the paper as a potential limitation, but it is important to explain what was used for this study in the manuscript.  

RESPONSE 12: We expanded this section on page 4 to add information that suggested by the reviewer.

Point 13: Good idea to include the actual, WHO issued preventive behaviors in the survey.

RESPONSE 13: We thank for the reviewer’s comment.

Point 14: For each potential mediation variable, please include the total number of items used or specify if the measure was a single item.

RESPONSE 14: We added that information for each potential mediation variable.

Point 15: What were the five response options for the variety of interpersonal information sources? Never to very often is listed in the measure, but later, the authors indicate that “often” and “always” were combined to reflect frequent users. Please clarify and add this information to the manuscript.

RESPONSE 15: We added the five response options on page 5.

Point 16: What was the rationale for dichotomizing the variety of interpersonal information sources?

RESPONSE 16: The quality of health information from different source varies. Instead of focusing on how frequently people use each source for information about COVID-19, we focused on the variety of source use because checking various sources to confirm information and address discrepancies is a recommended strategy to ensure using accurate information and make appropriate health decision. We added this rationale on page 5.

Point 17: How were media sources evaluated by participants? Please clarify and add that information to the manuscript.

RESPONSE 17: We listed the six items assessing participants’ information appraisal on page 6.

Point 18: Discussion: I would like to see more of an explanation as to why rural residents have poorer information appraisal skills than urban residents and their implications. In other words, I’m asking the authors to discuss potential reasons for the discrepancies in information appraisal between rural and urban residents and their implications for rural communities during the pandemic.

RESPONSE 18: We added the explanation and intervention strategies to improve information appraisal skills targeting rural populations on page 11.

Reviewer 3 Report

I think the authors present an interesting and important topic. They make a good work in providing motivation and background to their results and conclusiones. The article requires some minor clarifications.

I attach some comments. 

Author Response

Dear Reviewer 3,

We would like to take this opportunity to thank you for the constructive feedback and the opportunity to make revisions to our manuscript. The comments were very helpful as we revised sections of our paper and has resulted in a stronger contribution to the literature. Below, we address each comment in turn, and indicate where in the manuscript text to find specific revisions and newly added information (with tracking changes).

Point 1: Line 39: What does health behavior refer?

RESPONSE 1: On page 1, we clarified the term “preventive health behaviors” and added a definition of it.

Point 2: Line 100, 107, 108: Provide more details about the sampling procedure and add descriptive statistics in text.

RESPONSE 2: On page 3, we added the number of study invitations SoJump sent out (n = 1,717) and the number of respondents who completed the survey (n = 1,692). We also provided more descriptive statistics about sample demographics.

Point 3: Line 117: Are these rural or just smaller? What other feature, besides population size, is considered to classify a unit as "rural"?;  Line 124: OK, but some of them are expected to be already less prevalent in rural settings before COVID-19, as you previously stated.

RESPONSE 3: We agree with the reviewer that previous research indicate rural-urban disparities in preventive health behaviors related to diet and physical activity. People might change their behaviors during a pandemic. Therefore, our survey included eight types of preventive behaviors from the WHO COVID-19 prevention guidebook.

Point 4: Line 175: You should add a section on path analysis, since this is the methodology used for data analysis.

RESPONSE 4:  On page 4, we added subheadings for “multiple linear regressions” and “path analysis”.

Point 5: Line 298: Anything on the sample and the chosen methodology?

RESPONSE 5: In the limitation section, we added the disadvantages of our sampling method.

Round 2

Reviewer 1 Report

I think the authors have addressed my comments and suggestions. There are grammatical and mechanical errors in the manuscript and some awkward sentences here and there. They need to be cleaned up with careful proofreading and editing.